# γ-Aminobutyric Acid Priming Alleviates Acid-Aluminum Toxicity to Creeping Bentgrass by Regulating Metabolic Homeostasis

**DOI:** 10.3390/ijms241814309

**Published:** 2023-09-20

**Authors:** Min Zhou, Yan Yuan, Junnan Lin, Long Lin, Jianzhen Zhou, Zhou Li

**Affiliations:** College of Grassland Science and Technology, Sichuan Agricultural University, Chengdu 611130, China; 2021202098@stu.sicau.edu.cn (M.Z.); 2021302105@stu.sicau.edu.cn (Y.Y.); 2021202092@stu.sicau.edu.cn (L.L.); 2020302091@stu.sicau.edu.cn (J.Z.)

**Keywords:** aluminum toxicity, photochemical efficiency, organic acid, osmotic adjustment, oxidative damage, gene expression

## Abstract

Aluminum (Al) toxicity is a major limiting factor for plant growth and crop production in acidic soils. This study aims to investigate the effects of γ-aminobutyric acid (GABA) priming on mitigating acid-Al toxicity to creeping bentgrass (*Agrostis stolonifera*) associated with changes in plant growth, photosynthetic parameters, antioxidant defense, key metabolites, and genes related to organic acids metabolism. Thirty-seven-old plants were primed with or without 0.5 mM GABA for three days and then subjected to acid-Al stress (5 mmol/L AlCl_3_·6H_2_O, pH 4.35) for fifteen days. The results showed that acid-Al stress significantly increased the accumulation of Al and also restricted aboveground and underground growths, photosynthesis, photochemical efficiency, and osmotic balance, which could be effectively alleviated by GABA priming. The application of GABA significantly activated antioxidant enzymes, including superoxide dismutase, peroxidase, catalase, and ascorbate peroxidase, to reduce oxidative damage to cells under acid-Al stress. Metabolomics analysis demonstrated that the GABA pretreatment significantly induced the accumulation of many metabolites such as quinic acid, pyruvic acid, shikimic acid, glycine, threonine, erythrose, glucose-6-phosphate, galactose, kestose, threitol, ribitol, glycerol, putrescine, galactinol, and myo-inositol associated with osmotic, antioxidant, and metabolic homeostases under acid-Al stress. In addition, the GABA priming significantly up-regulated genes related to the transportation of malic acid and citric acid in leaves in response to acid-Al stress. Current findings indicated GABA-induced tolerance to acid-Al stress in relation to scavenging of reactive oxygen species, osmotic adjustment, and accumulation and transport of organic metabolites in leaves. Exogenous GABA priming could improve the phytoremediation potential of perennial creeping bentgrass for the restoration of Al-contaminated soils.

## 1. Introduction

Acidic soils are often accompanied by aluminum (Al) toxicity, which inhibits plant growth and development, resulting in reduced crop yield and quality worldwide, especially in heavy rainfall regions. In recent years, the area and degree of soil acidification have increased further because of frequent acidic rain and continuous industrial pollution [1,2,3]. It has been widely reported that acid-Al stress significantly inhibited aboveground growth, gas exchange, and photosynthesis [4,5,6,7,8,9]. Plants have evolved several mechanisms, such as an antioxidant defense system to defend against the toxic effects of acid-Al stress which lead to increased production of reactive oxygen species (ROS) in cells [10]. Stress-induced overaccumulation of ROS, such as superoxide anion (O_2_^∙−^) and hydrogen peroxide (H_2_O_2_), oxidizes membrane systems and also destroys protein and organelle structure. However, oxidative damage can be relieved by various antioxidant enzymes, including superoxide dismutase (SOD), peroxidase (POD), catalase (CAT), ascorbate peroxidase (APX), etc. [11]. A previous study by Zheng et al. found that nitric oxide could activate POD and CAT to scavenge ROS, thereby protecting watermelon (*Citrullus lanatus*) plants from acid-Al stress [12]. In addition, an exogenous supply of a suitable concentration of boron has been proven to be a good approach to improving acid-Al tolerance by regulating the antioxidant defense system [13]. 

Except for the antioxidant system, the biosynthesis and transport of organic acids also play necessary roles in acid-Al detoxification. Intermediate metabolites of the tricarboxylic acid (TCA) cycle, such as malic acid and citric acid, can combine with Al^3+^ to form a complex of Al–organic acids which could be isolated into vacuoles to maintain low levels of free Al^3+^ in the cytoplasm [14,15]. *CSs* or *MDHs* encoding citrate synthases or malate dehydrogenases are responsible for the biosynthesis of citric acid or malic acid in plants, respectively [16,17]. Multidrug and toxic compound extrusions (MATEs) or aluminum-activated malate transporters (ALMTs) are involved in the transport of malic acid or citric acid, respectively [18,19]. The overexpression of *BraMDH* had a positive influence on acid-Al tolerance in *Arabidopsis thaliana* [20]. Transgenic canola (*Brassica napus*) plants overexpressing a *CS* gene exhibited significantly enhanced tolerance to acid-Al stress associated with higher activities of CS and MDH in the TCA cycle [21]. Moreover, the overexpression of *GsMATE* that was cloned from soybean (*Glycine soja*) could improve the tolerance to Al toxicity in *Arabidopsis thaliana* [22]. In addition, it has been found that sensitivity to proton rhizotoxicity 1 (STOP1) was a key regulator of Al tolerance through the pathway of organic metabolism because it directly regulated the expression of *MATE* [23] or *ALMT1* in plants [24]. *Arabidopsis thaliana stop1* mutant was hypersensitive to Al^3+^ toxicity due to a lack of Al-induced *AtALMT1* expression [25]. These previous studies indicated that the regulation of accumulation and transportation of organic acids are of primary importance for plant adaptation to acid-Al environments.

As a nonprotein amino acid, γ-aminobutyric acid (GABA) has been widely studied because of its positive functions of signaling transduction, antioxidant, osmotic adjustment (OA), energy metabolism, and carbon-nitrogen balance in plants suffering from multiple abiotic stresses [26,27]. It has been found that acid-Al stress significantly up-regulated glutamate decarboxylase activity which is a key enzyme for endogenous GABA biosynthesis, and exogenous application of GABA could effectively mitigate Al toxicity to *Liriodendron chinense* × *tulipifera* plants in relation to significant improvements in *MATE1/2* expression and antioxidant capacity for ROS scavenging [28]. However, it is not fully understood how GABA regulates metabolic homeostasis, especially for malic and citric acids metabolism against acid-Al stress. As a gramineous turfgrass with an appropriate pH range from 5.0 to 6.5 for growth and development, creeping bentgrass (*Agrostis stolonifera*) is utilized worldwide in urban green spaces and sports grounds such as golf courses, lawn tennis, or bowling greens, but acid-Al stress significantly decreases its growth and turf quality [29]. Objectives of the current study focused on investigating the effects of GABA priming on plant growth, photosynthetic response, and antioxidant defense and further revealed the GABA-induced metabolic homeostasis in creeping bentgrass based on analyses of metabolome and key genes involved in biosynthesis and transport of malic and citric acids under acid-Al stress.

## 2. Results

### 2.1. Effects of GABA Priming on Aluminum Content and Plant Growth under Normal Conditions and Acid-Al Stress

As shown in Figure 1A, there was a significant morphological difference between the plants grown under normal conditions and the plants when subjected to acid-Al stress. Acid-Al stress led to a significant accumulation of Al in leaves, but GABA priming significantly inhibited Al accumulation induced by the acid-Al stress (Figure 1B). Relative growth rate (RGR) of leaf and root significantly declined in both GABA-primed and non-primed plants, and the RGR of shoot also significantly decreased in plants without GABA priming under acid-Al stress (Figure 1C). However, acid-Al stress did not significantly affect the RGR of shoots in the GABA-primed plants (Figure 1C). GABA-primed plants exhibited a 21.52%, 30.16%, or 88.91% increase in RGR of leaf, shoot, or root than the plants without GABA priming under acid-Al stress (Figure 1C). Shoot length was significantly inhibited by acid-Al stress in all plants, and GABA priming not only alleviated the inhibitory effect of acid-Al on shoot length but also promoted shoot length under normal conditions (Figure 1D).

### 2.2. Effects of GABA Priming on Leaf Water Status and Photosynthesis under Normal Conditions and Acid-Al Stress

Under normal conditions, exogenous GABA pretreatment had no significant effect on relative water content (RWC) in leaves. After 15 days of acid-Al stress, the RWC significantly decreased in all treatments, but as compared to these plants without GABA priming under acid-Al stress, RWC increased by 25.37% after the GABA priming (Figure 2A). Acid-Al significantly decreased osmotic potential (OP) in GABA-primed plants, whereas it improved the OP in the plants without GABA priming under acid-Al stress (Figure 2B). Contents of chlorophyll a (Chl a), Chl b, and total Chl, as well as the ratio of Chl a/b, were significantly reduced under acid-Al stress. Chl a, Chl b, total Chl content, or Chl a/b in leaves treated with GABA was 41.25%, 29.67%, 37.64%, or 16.11% higher than those plants without GABA priming under acid-Al stress, respectively (Figure 3A). GABA priming did not significantly affect intercellular carbon dioxide concentration (Ci), stomatal conductance (Gs), net photosynthetic rate (Pn), transpiration rate (Tr), and water use efficiency (WUE) in leaves under normal conditions (Figure 3B). In response to acid-Al stress, Ci significantly increased, while Gs, Pn, Tr, and WUE significantly decreased in all treatments. GABA priming effectively mitigated the acid-Al-induced increase in Ci and declines in Gs, Pn, Tr, and WUE (Figure 3B). Acid-Al stress led to significant reductions in photochemical efficiency (Fv/Fm) and performance index on absorption basis (PI_ABS_). However, the acid-Al+GABA treatment had a 21.0% or 228.84% increase in Fv/Fm or PI_ABS_ than the acid-Al treatment, respectively (Figure 3C,D).

### 2.3. Effects of GABA Priming on Antioxidant Enzyme Activities and Oxidative Damage under Normal Conditions and Acid-Al Stress

Activities of SOD, POD, CAT, and APX in leaves were not significantly affected by GABA priming under normal conditions (Figure 4A–D). SOD and POD activities decreased in plants without GABA priming when subjected to acid-Al stress, but their activities in GABA-primed plants were maintained at normal levels under acid-Al stress (Figure 4A,B). In addition, acid-Al stress did not activate CAT and APX activities in non-primed plants but significantly up-regulated their activities in GABA-pretreated plants (Figure 4C,D). GABA pretreatment significantly increased SOD, POD, CAT, or APX activity by 64.21%, 75.59%, 52.22%, or 32.85% under acid-Al stress, respectively (Figure 4A–D). Acid-Al significantly induced the accumulation of malondialdehyde (MDA) in plants without GABA priming, but this symptom was not observed in GABA-primed plants (Figure 4E). Electrolyte leakage (EL) significantly increased in leaves of all treatments under acid-Al stress (Figure 4F). As compared to the acid-Al treatment, the acid-Al+GABA treatment exhibited an 87.36% decline in EL (Figure 4F).

### 2.4. Effects of GABA Priming on Metabolites Profile under Normal Conditions and Acid-Al Stress

A total of 73 metabolites were identified and quantified in leaves (Figure 5A), and specific information about those metabolites, including 23 organic acids, 18 amino acids, 16 sugars, and 16 other metabolites, are shown in Appendix A. GABA priming significantly reduced the accumulation of total organic acids, amino acids, sugars, or other metabolites under normal conditions, but did not affect their accumulations under acid-Al stress (Figure 5B). Four groups (C+GABA vs. C, acid-Al+GABA vs. acid-Al, acid-Al vs. C, and acid-Al+GABA vs. C) were set to compare the effects of GABA priming or acid-Al stress in leaves. Only 1.35% of metabolites were up-regulated in the C+GABA vs. C (Figure 5C). A 33.78% of metabolites were up-regulated in the acid-Al+GABA vs. acid-Al, whereas an 89.19% of metabolites were down-regulated in both acid-Al vs. C and acid-Al+GABA vs. C (Figure 5C). Nineteen metabolites were significantly affected by GABA priming or acid-Al stress in leaves (Figure 6A–D). Acid-Al stress down-regulated accumulations of malic acid, citric acid, quinic acid, shikimic acid, and pyruvic acid in all treatments, but the acid-Al+GABA treatment maintained significantly higher contents of quinic acid, shikimic acid, and pyruvic acid than the acid-Al treatment (Figure 6A). Acid-Al stress increased the accumulation of proline, but GABA priming significantly mitigated the acid-Al-induced accumulation of proline in leaves (Figure 6B). The acid-Al+GABA exhibited significantly higher contents of glycine, threonine, and cycloleucine than the acid-Al treatment (Figure 6B). GABA-primed plants also accumulated more erythrose, glucose-6-phosphate, galactose, kestose, threitol, ribitol, glycerol, putrescine (Put), galactinol, and myo-inositol than those plants without GABA priming under acid-Al stress (Figure 6C,D).

### 2.5. Effects of GABA Priming on Expression Levels of Genes Involved in Accumulation and Transport of Malic and Citric Acids under Normal Conditions and Acid-Al Stress

GABA priming did not significantly induce changes in expression levels of genes involved in the accumulation and transport of citric and malic acids under normal conditions (Figure 7A,B). The expression level of *ALMT9-like* or *mMDH-like* only significantly increased in the acid-Al+GABA or acid-Al treatment as compared to the C treatment, respectively (Figure 7A). Acid-Al stress induced *STOP1-like*, *cMDH-like*, *3-cIPMDH 2-like*, and *CS-like* expressions in leaves. A significantly higher expression level of *STOP1-like* was detected in the acid-Al+GABA than that in the acid-Al treatment (Figure 7A). Acid-Al stress significantly increased expression levels of *MATE14-like*, *MATE19-like*, *MATE27-like*, *MATE29-like*, and *MATE48-like* (Figure 7B). The acid-Al+GABA exhibited a 42.14%, 49.08%, or 43.31% increase in transcript levels of *MATE19-like*, *MATE27-like*, or *MATE29-like* than the acid-Al treatment, respectively, but there were no significant differences in expression levels of *CS-like*, *MATE14-like,* and *MATE48-like* between the acid-Al and acid-Al+GABA (Figure 7B). Figure 8 shows the potential mechanism of GABA-regulated tolerance to acid-Al stress associated with growth, photosynthetic, antioxidant, and metabolic homeostases in leaves.

## 3. Discussion

### 3.1. GABA-Regulated Tolerance to Acid-Al Stress in Relation to Changes in Al Accumulation, Photosynthesis, and Osmotic Balance

After being subjected to acid-Al stress, roots were compelled to uptake redundant Al^3+^, which were further transferred to aboveground tissues, resulting in Al toxicity to plants [30]. Overaccumulation of Al interferes with plant growth and metabolic processes, including ROS balance, carbon and nitrogen metabolisms, photosynthesis, etc. It has been reported that acid-Al stress significantly restrained elongation and dry mass accumulation of roots and shoots and also reduced RWC, Chl content, Pn, and WUE in leaves of apple (*Malus pumila*) or barley (*Hordeum vulgare*) seedlings [31,32]. As compared to Al-sensitive maize variety BRS1010, Al-tolerant BRS1055 accumulated less Al in shoots, accompanied by a significantly higher RGR of shoots under acid-Al stress [33]. Similar findings were obtained in the current study, which showed that acid-Al-induced growth retardant, Al enrichment, and declines in Chl content, Pn, and photochemical efficiency could be significantly mitigated by the GABA priming in creeping bentgrass plants. This indicated the positive effects of GABA priming on creeping bentgrass under acid-Al stress. An earlier study by Silambarasan et al. also found that the beneficial role of rhizobacterial *Curtobacterium herbarum* strain CAH5 in alleviating acid-Al toxicity to *Lactuca sativa* plants was related to the inhibition of Al accumulation in shoots [34]. Interestingly, significantly reduced Gs along with an elevatory Ci were observed in leaves of creeping bentgrass in response to a long period of acid-Al stress, indicating non-stomatal limitation due to impaired chloroplast could be the main limiting factor to photosynthesis. The exogenous application of GABA was conducive to the alleviation of stress-induced Chl degradation and reduced photosynthetic performance, which has been proved in many previous studies as well as current findings [35,36,37,38]. In addition, acid-Al toxicity also caused physiological drought, as demonstrated by a significant decline in leaf RWC [39]. Maintenances of high WUE and low OP have been recognized as two key strategies for plants tolerating water-deficit conditions [40,41,42]. The GABA priming significantly mitigated physiological drought induced by acid-Al stress by maintaining high WUE and OA in creeping bentgrass.

### 3.2. GABA-Regulated Tolerance to Acid-Al Stress in Relation to Changes in Oxidative Damage and Antioxidant Defense

Acid-Al toxicity triggers ROS burst, resulting in oxidative damage to cell membrane systems, nucleotides, chloroplasts, and other cytoplasmic organoids [43]. Membrane peroxidation and cell membrane stability have been widely used to evaluate stress-caused oxidative damage in plants, as mainly reflected by the accumulation of MDA and high EL levels [44,45,46]. SOD is responsible for O_2_^∙−^ scavenging, which is known as the first line of antioxidant defense. POD, CAT, and APX exhibit similar functions in the removal of H_2_O_2_ from cells [47,48]. Acid-Al stress led to overaccumulation of MDA and an increase in EL of peanut (*Arachis hypogaea*) plants. However, SOD, POD, and CAT were significantly activated by acid-Al stress to provide relief from oxidative damage [49]. Similarly, cowpea (*Vigna unguiculata*) plants could activate the SOD, POD, CAT, and APX in response to a high concentration of acid-Al (10 mM) stress, thereby alleviating membrane peroxidation and improving cell membrane stability [30]. Beneficial effects of GABA on antioxidant enzyme activities for ROS homeostasis have been reported under various abiotic stresses such as salt stress [50], alkali stress [51], and low light [52]. A recent study by Wang et al. also found that MDA content continuously increased in hybrid *Liriodendron* plants when subjected to 30 μM Al stress for three days, but exogenous application of GABA could further improve Al-induced increases in antioxidant enzyme activities [28]. Although a high concentration of Al (5 mM) stress failed to activate the antioxidant defense system, the GABA priming up-regulated activities of SOD, POD, CAT, and APX in leaves of creeping bentgrass after 15 days of acid-Al stress. These findings indicated that the antioxidant response varied from one plant species to another species depending on the duration and intensity of acid-Al stress, and the GABA exhibited positive effects on antioxidant enzyme activities and ROS scavenging during a prolonged period of acid-Al stress.

### 3.3. GABA-Regulated Tolerance to Acid-Al Stress in Relation to Changes in Accumulation and Transport of Organic Acids

Accumulations of organic metabolites such as organic acids, amino acids, and sugars are another important adaptive strategy for acid-Al tolerance in plants since multiple roles of these metabolites in OA, antioxidant, and Al chelation [15,53,54]. However, severe stress often causes metabolite deficits due to loss of photosynthetic capacity [55]. Current results showed that acid-Al stress significantly decreased the accumulation of various metabolites in leaves of creeping bentgrass in all treatments, while GABA-primed creeping bentgrass maintained higher contents of quinic acid, shikimic acid, and pyruvic acid than non-primed plants after being exposed to a long term of acid-Al stress. Quinic acid is widely found in plants and known for its important biological properties in medicinal plants, such as antioxidant activity [56,57]. Pyruvic acid has the primary role in the biosynthesis and transformation of organic and amino acids in plants [58,59]. Differential metabolic responses to acid-Al stress were found in two tea (*Camellia sinensis*) cultivars, JHC and YS. Less Al was accumulated in roots and shoots of the YS which mainly up-regulated quinic and shikimic pathways, but the JHC which enriched more Al in whole plants improved the accumulation of pyruvic acid to adapt to acid-Al stress [60]. The specific function of quinic acid, shikimic acid, or pyruvic acid on acid-Al tolerance in creeping bentgrass and other plant species still needs to be further investigated in the future.

However, the GABA priming further decreased accumulations of malic and citric acids in leaves of creeping bentgrass under acid-Al stress. Generally, improvements in accumulations and transports of malic acid and citric acid are beneficial to acid-Al tolerance because these organic acids can combine with Al^3+^ to produce Al oxalate for detoxification of Al in the cytoplasm [61]. The analysis of gene expression demonstrated that the GABA priming significantly up-regulated *STOP1-like* and many genes involved in the transport of malic acid (*ALMT9-like*) or citric acid (*MATE19-like*, *MATE27-like*, and *MATE29-like*) in creeping bentgrass under acid-Al stress. ALMT9 is located in the vacuole membrane and is responsible for the transportation of malate from the cytoplasm into the vacuole. The abundance of *ALMT9* increased significantly in tomato (*Solanum lycopersicum*) plants after acid-Al treatment associated with elevated malic acid transport and enhanced acid-Al tolerance [62]. The MATE family is an important group of multidrug efflux transporters that transport a broad range of organic compounds such as citric acid into vacuole or rhizosphere. Ribeiro et al. overexpressed a *SbMATE* gene in sugarcane (*Saccharum officinarum*), which could effectively improve tolerance to acid-Al [63]. As a key transcription factor for regulating *ALMT* and *MATE* genes, the function of *STOP1*-regulated acid-Al tolerance has been widely elucidated in plants. For instance, the overexpression of *SbSTOP1d* cloned from sweet sorghum (*Sorghum bicolor*) could rescue *Arabidopsis thaliana atstop1* mutant from the hypersensitivity to Al stress [64]. A *SISTOP1* activated *MATE* pathway to mediate acid-Al tolerance of tomato [65]. Our current results indicated that the GABA priming could mainly improve transports of malic and citric acids in leaves of creeping bentgrass when subjected to acid-Al stress instead of their accumulation.

### 3.4. GABA-Regulated Tolerance to Acid-Al Stress in Relation to Changes in Accumulations of Amino Acids and Other Metabolites

Although acid-Al stress reduced accumulations of most of the metabolites in creeping bentgrass plants, the GABA-primed plants maintained higher contents of many metabolites than non-primed ones under acid-Al stress, such as glycine, threonine, erythrose, glucose-6-phosphate, galactose, kestose, threitol, ribitol, glycerol, Put, galactinol, and myo-inositol. These amino acids, sugars, and other metabolites play multiple positive roles in tolerance to abiotic stress as osmotic regulators, antioxidants, stress-defensive signaling regulators, or intermediates for energy metabolism in plants [66,67,68,69,70,71,72,73,74]. In particular, the function of Put was involved in the mitigation of Al-induced inhibition of wheat (*Triticum aestivum*) root growth by reducing ethylene production [75]. On the contrary, the exogenous application of Put biosynthetic inhibitor aggravated Al-induced oxidative damage to wheat roots [76]. In addition, the application of a suitable dose of Put (1 mM) improved antioxidant response to acid-Al stress, thereby alleviating growth retardance and oxidative damage to salvinia (*Salvinia natans*) plants [77]. Myo-inositol regulates cellular signaling pathways, including the responses to stress and ascorbic acid biosynthesis [72]. Exogenous myo-inositol could alleviate salt stress by enhancing antioxidants, membrane stability, and upregulation of stress-responsive genes [78]. Moreover, when myo-inositol synthesis decreases, plants would be sensitive to stress or accelerate death [79,80]. Thus, these results indicated that the GABA priming regulated acid-Al tolerance of creeping bentgrass by maintaining accumulation and homeostasis of key metabolic metabolites such as Put and myo-inositol in leaves in favor of aboveground growth and stress defense.

## 4. Materials and Methods

### 4.1. Plant Materials and Growth Conditions

Creeping bentgrass cultivar ‘Penncross’ was used for the experiment. Seeds (5 g/m^2^) were sown evenly in a rectangular container (25 cm length, 15 cm width, and 15 cm height) that was filled with quartz sands. Seeds first germinated in distilled water for 7 days, and then the seedlings were cultivated in half-strength Hoagland nutrient solution for 30 days [81]. All containers were placed randomly in controlled growth chambers (a 24 h day-night cycle of 14 h at 23 °C in 700 μmol·m^−2^·s^−1^ PAR light and 12 h at 19 °C in darkness), and the position of each container was shifted every day to eliminate spatial effect. All plants were then separated into two groups: one group was cultivated in standardized Hoagland nutrient solution for 3 days as the non-GABA treatment; another group was cultivated in the Hoagland nutrient solution containing 0.5 mmol/L GABA for 2 days as the GABA pretreatment [37]. GABA-treated and non-treated plants were further divided into four treatments: (1) C [control, plants without GABA priming grew in standardized Hoagland nutrient solution (pH 6.20) for 15 days]; (2) C+GABA [control plus GABA, plants with GABA priming grew in standardized Hoagland nutrient solution (pH 6.20) for 15 days]; (3) acid-Al [acid-Al stress, plants without GABA priming grew in the Hoagland nutrient solution (pH 4.35) containing 5 mmol/L AlCl_3_·6H_2_O for 15 days]; and (4) acid-Al+GABA [acid-Al stress plus GABA, plants with GABA priming grew in the Hoagland nutrient solution (pH 4.35) containing 5 mmol/L AlCl_3_·6H_2_O for 15 days]. Six independent biological replications (six containers) were used for each treatment, and the Hoagland nutrient solution was refreshed every day to avoid significant changes in solution concentration and pH value.

### 4.2. Measurements of Growth, Water Status, and Aluminum Content

The calculation formula of RGR is the RGR = (lnW_f_ − lnW_i_)/Δ*_t_*, where W_i_ and W_f_ represent initial (after GABA priming and before acid-Al stress) and final (after 15 days of acid-Al stress) dry weights of plants, respectively, and Δ*_t_* is the interval time (d) between the two measurements [82]. The Al content in the leaves was detected using ICP-MS. Fresh leaves were oven-dried to a consistent weight and then ground into fine powders. A total of 0.1 g of powders, 5 mL of concentrated nitric acid, and 2 mL of hydrogen peroxide were mixed in the digestion tank until all powders were digested completely. Digest samples were used for the determination of Al content. For RWC, the fresh weight (FW) of leaves was weighed, and then these leaves were immersed in 10 mL of distilled water for 24 h to detect turgid weight (TW). After that, the leaves were oven-dried at 75 °C to a constant dry weight (DW). The RWC was calculated according to the formula RWC (%) = (FW − DW)/(TW − DW) × 100% [83]. For the determination of OP, the 0.2 g of leaves were completely soaked in deionized water for 12 h. These leaves were removed from the water, and the surface moisture was wiped off. The leaf juice was squeezed out, and 10 μL of them was put into the osmometer (WESCOR-5600, Logan, UT, USA) to measure the osmolality (c), and then OP (Mpa) was calculated by using the formula C × 2.58 × 10^−3^ [84].

### 4.3. Measurements of Chlorophyll Content and Photosynthetic Parameters

The 0.1 g of fresh leaves were cut from plants and placed in a 15 mL centrifuge tube which was filled with 10 mL of dimethyl sulfoxide (DMSO). These leaves were completely immersed in the DMSO solution, and tubes were placed in the dark until all leaves turned completely white. The absorbance of the extraction solution was detected by using a spectrophotometer (Spectronic Instruments, Rochester, NY, USA) at 645 nm and 663 nm [85]. Leaves were subjected to dark adaptation for 30 min, and the Fv/Fm and PI_ABS_ were recorded using a Chl fluorometer (Pocket PEA) [86]. Portable photosynthetic system CIRAS-3 was required to detect Pn, Tr, WUE, Ci, and Gs. A single layer of leaves overspread leaf chamber which provided the stable 400 mL/L CO_2_ and 800 mmol photon m^−2^ red and blue light [87].

### 4.4. Measurements of Antioxidant Enzyme Activity, Oxidative Damage, and Cell Membrane Stability

The 2 mL of 4 °C phosphate buffer (150 mM and pH 7.0) was mixed with 0.1 g of fresh sample, and the mixture was homogenated in a refiner. The homogenate was then centrifugated at 10,000× *g* for 10 min. The supernatant was used for the determination of antioxidant enzyme activities, including SOD, POD, CAT, and APX, as well as MDA content. The riboflavin–nitrotetrazolium blue chloride (NBT) method was used to determine SOD activity at 560 nm [88], and the absorbance of CAT or POD activity was measured at 240 nm or 470 nm, respectively [89]. The activity of APX was determined at 290 nm according to the method of Nakano and Asada [90]. MDA content was measured by the thiobarbituric acid method, and the absorbance of the reaction solution was detected at 532 nm and 600 nm, respectively [91]. Fresh leaves (0.1 g) were soaked in 15 mL of deionized water at 20 °C for 24 h, and the initial electrolyte permeability (E_i_) of the solution was detected by a conductivity meter. After being autoclaved at 120 °C for 20 min and cooled down to room temperature, the final electrolyte permeability (E_f_) of the solution was detected. EL was calculated by using the formula EL = E_i_/E_f_ × 100% [92].

### 4.5. Determination of Metabolomics and Gene Expression Analysis

Freeze-dried leaf samples were ground into fine powder after being lyophilized by using a lyophilizer (LGJ-10C, Chengdu, China), and a total of 20 mg of each sample was used to extract the total metabolites by using the method from our previous research [93]. Metabolites analysis was conducted by gas chromatography-mass spectrometry (GC-MS) [94], and identification was conducted by the Chroma TOF software (v. 4.50.8.0, LECO, St. Joseph, MI, USA) and commercial compound libraries NIST 2005 (PerkinElmer Inc., Waltham, MA, USA) and Wiley 7.0 (John Wiley and Sons Ltd., Hoboken, NJ, USA).

For the analysis of gene expression level, total RNA was extracted within 0.1 g of fresh leaf samples using a HiPure Universal RNA kit (Magen), and these RNAs were then reversed into cDNA using a reverse transcription kit (MonScriptTM RTIII All-in-one Mix with dsDNase kit from Monad). The reaction system of real-time quantitative fluorescent PCR (qRT-PCR) included a 5 μL Taq SYBR Green qPCR Premix, a 0.5 μL forward primer, a 0.5 μL reverse primer, a 1 μL cDNA, and a 3 μL Nuclease-Free water. PCR procedure included the following steps: pre-deformation for 30 s at 95 °C, deformation for 10 s at 95 °C, annealing at 55–65 °C for 10 s, and extending at 72 °C for 30 s (40 cycles from step 2 to step 4). Primer sequences and particular annealing temperatures of tested genes are shown in Appendix A. The formula for calculating the relative expression of genes was obtained from the study of Livak and Schmittgen [95].

### 4.6. Statistical Analysis

Statistical analysis was conducted by using *SPSS 26.0* based on the Tukey test (IBM, Armonk, NY, USA). The least significance test (LSD) and T-test were used to test the variance of different treatments *at p* ≤ 0.05.

## 5. Conclusions

In conclusion, acid-Al toxicity seriously restricted plant growth and photosynthesis. However, GABA pretreatment could significantly mitigate the detrimental effects of acid-Al stress on RGR, Chl content, Fv/Fm, PI_ABS_, Gs, Pn, Tr, and WUE in leaves. The application of GABA also played a positive role in improving antioxidant enzyme activities to reduce oxidative damage to cells. In addition, metabolomic data demonstrated that the GABA pretreatment induced the accumulation of many metabolic substances such as quinic acid, pyruvic acid, shikimic acid, glycine, threonine, erythrose, glucose-6-phosphate, galactose, kestose, threitol, ribitol, glycerol, Put, galactinol, and myo-inositol associated with osmotic, antioxidant, and metabolic homeostases under acid-Al stress. It is noteworthy that genes related to the transport of malic and citric acids were up-regulated by the GABA priming in response to acid-Al stress. Current findings enrich regulatory mechanisms of GABA-induced tolerance to acid-Al stress in perennial grass species, but transcriptional and post-transcriptional regulations still need to be further studied in the future.

## Figures and Tables

**Figure 1 ijms-24-14309-f001:**
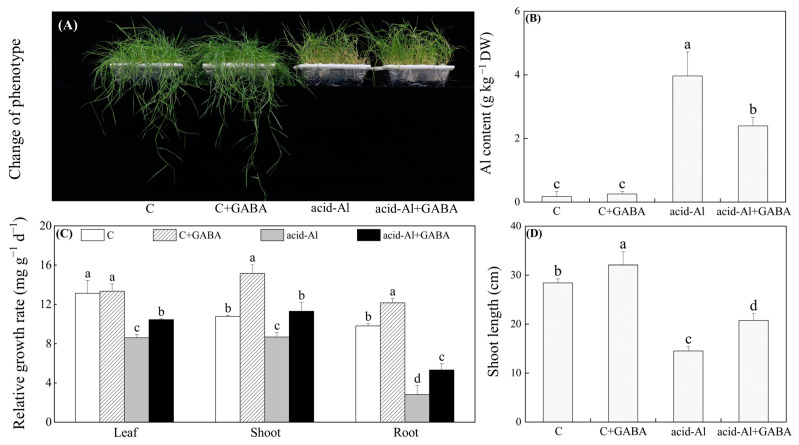
Effects of GABA priming on (**A**) phenotypic changes, (**B**) aluminum content, (**C**) relative growth rate, and (**D**) shoot length of creeping bentgrass under normal conditions and acid-aluminum stress. Vertical bars above columns indicate standard errors of means (*n* = 3), and different letters indicate significant differences among four treatments based on the LSD test (*p* ≤ 0.05). C—control; C+GABA—control+GABA; acid-Al—acid-Al stress; acid-Al+GABA—acid-Al stress+GABA.

**Figure 2 ijms-24-14309-f002:**
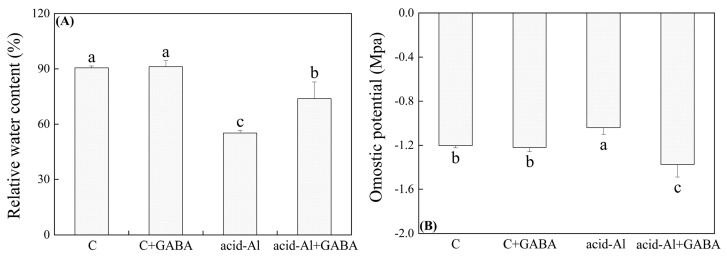
Effects of GABA priming on (**A**) relative water content and (**B**) osmotic potential in leaves of creeping bentgrass under normal conditions and acid-aluminum stress. Vertical bars above columns indicate standard errors of means (*n* = 3), and different letters indicate significant differences among four treatments based on the LSD test (*p* ≤ 0.05).

**Figure 3 ijms-24-14309-f003:**
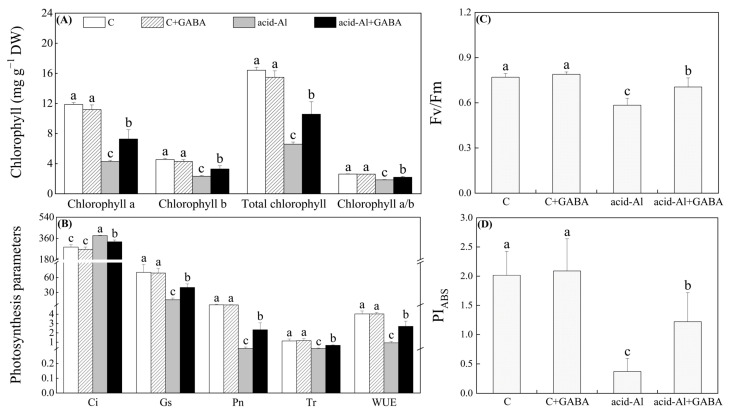
Effects of GABA priming on (**A**) chlorophyll, (**B**) photosynthetic parameters, (**C**) Fv/Fm, and (**D**) PI_ABS_ in leaves of creeping bentgrass under normal conditions and acid-aluminum stress. Vertical bars above columns indicate standard errors of means (*n* = 3), and different letters indicate significant differences among four treatments based on the LSD test (*p* ≤ 0.05).

**Figure 4 ijms-24-14309-f004:**
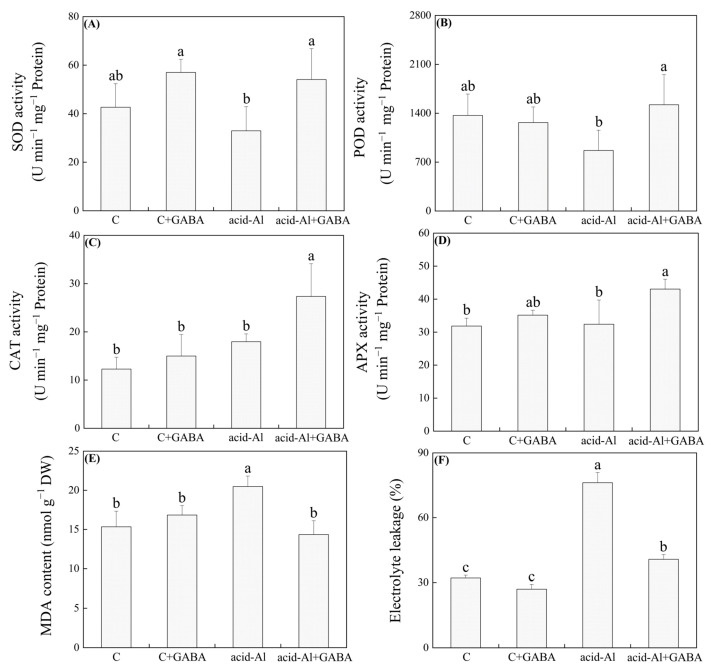
Effects of GABA priming on (**A**) SOD activity, (**B**) POD activity, (**C**) CAT activity, (**D**) APX activity, (**E**) MDA, and (**F**) EL in leaves of creeping bentgrass under normal conditions and acid-aluminum stress. Vertical bars above columns indicate standard errors of means (*n* = 3), and different letters indicate significant differences among four treatments based on the LSD test (*p* ≤ 0.05).

**Figure 5 ijms-24-14309-f005:**
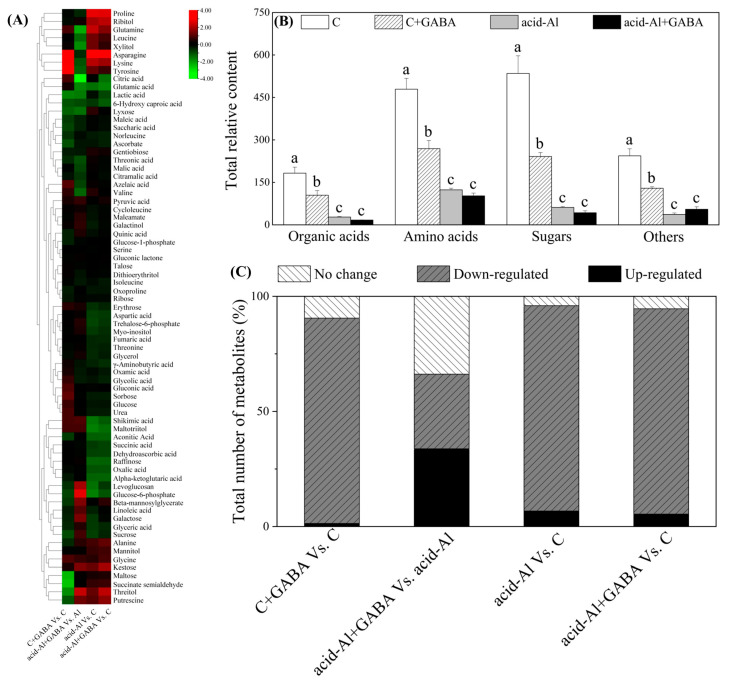
(**A**) Heat map of 73 metabolites in different comparable groups (red represents up-regulation and green represents down-regulation), (**B**) total contents of organic acids, amino acids, sugars, or other metabolites, and (**C**) the percentage of total number of metabolites in each comparable group in leaves of creeping bentgrass under normal condition and acid-aluminum stress. Vertical bars above columns indicate standard errors of means (*n* = 6), and different letters indicate significant differences among four treatments based on the LSD test (*p* ≤ 0.05).

**Figure 6 ijms-24-14309-f006:**
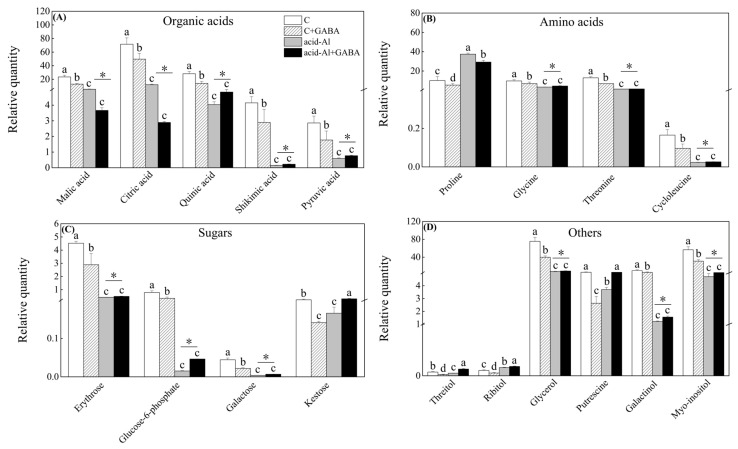
Effects of GABA priming on contents of (**A**) organic acids, (**B**) amino acids, (**C**) sugars, and (**D**) others in leaves of creeping bentgrass under normal conditions and acid-aluminum stress. Vertical bars above columns indicate standard errors of means (*n* = 6), and different letters indicate significant differences among four treatments based on the LSD test. The “*” represents a significant difference between the two treatments (Al and Al+GABA) (*p* ≤ 0.05).

**Figure 7 ijms-24-14309-f007:**
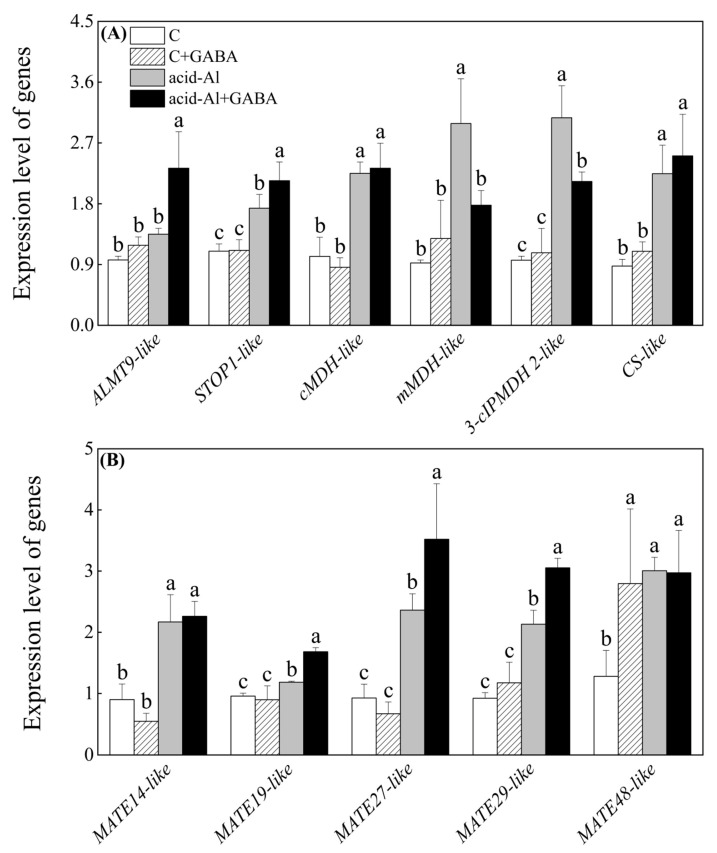
Effects of GABA priming on transcript levels of genes related to (**A**) malic and citric acids synthesis and transport and (**B**) citric acid transport in leaves of creeping bentgrass under normal conditions and acid-aluminum stress. Vertical bars above columns indicate standard errors of means (*n* = 3), and different letters indicate significant differences among four treatments based on the LSD test (*p* ≤ 0.05).

**Figure 8 ijms-24-14309-f008:**
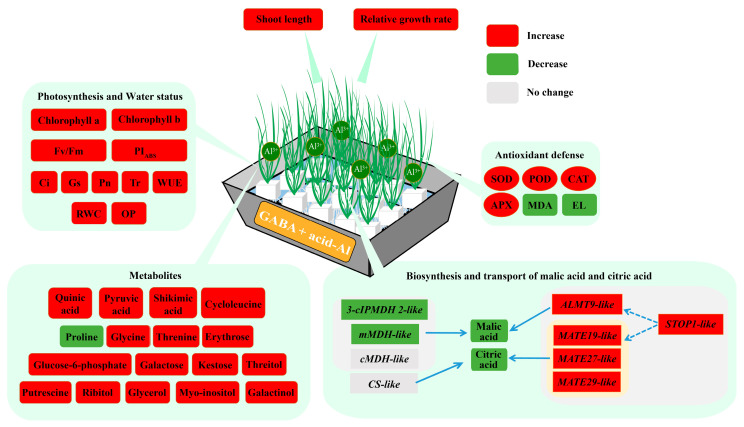
Potential mechanism regulated by GABA priming in leaves of creeping bentgrass under acid-aluminum stress. Results were demonstrated based on the comparable group acid-Al+GABA vs. acid-Al. Dashed lines represent that pathways still need to be further studied.

## Data Availability

Not applicable.

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
