# Peer review of "γ-Aminobutyric Acid Priming Alleviates Acid-Aluminum Toxicity to Creeping Bentgrass by Regulating Metabolic Homeostasis"

_ijms, 2023, doi:10.3390/ijms241814309_

Round 1

Reviewer 1 Report

Dear Editors,

Thank you for choosing me as a reviewer of the of the manuscript ID: ijms-2597039 entitled: ,, γ-Aminobutyric acid (GABA) priming alleviates acid-aluminum toxicity and growth retardation by maintaining photosynthetic, antioxidant, and metabolic homeostases in creeping bentgrassI hope that my comments will help authors to improve their manuscript.

Detailed remarks concerning the manuscript:

I suggest to modify the title of the manuscript to be concise and informative.

Abstract. Aluminium stress – the doses of stressor used in the should be mentioned.

The practical application of the results studies should be provided.

I suggest to divide ”Discussion” section into the subsections corresponded to the results section. The results of the study should be presented on the background of the literature data, not the other way around. The “Discussion” section should be slightly modified as some its parts sounds like introduction.

Materials and method. I suggest to include the subsection concerning statistical analysis.

References should be prepared strictly to the guidelines for authors. There are many editorial mistakes that should be improved. Mainly the latest information (bibliography from the latest five years). There is not possible to mention all of them. There are some examples:

a)                  All reference should contain full bibliographic data. See: “Hajiboland, R.; Panda, C. K.; Lastochkina, O.; Gavassi, M. A.; Habermann, G.; Pereira, J. F., Aluminum toxicity in plants: present and future. J Plant Growth Regul 2022.”

b)                 Once the full, but the other time abbreviated Journal names are provided.

c)                   all the species Latin names should be italicized.

d)                 ”Iuchi, S.; Koyama, H.; Iuchi, A.; Kobayashi, Y.; Kitabayashi, S.; Kobayashi, Y.; Ikka, T.; Hirayama, T.; Shinozaki, K.; Kobayashi, M., Zinc finger protein STOP1 is critical for proton tolerance in Arabidopsis and coregulates a key gene in aluminum tolerance. PNAS 2007, 104, (23), 9900-9905”. Arabidopsis should be italicized.

e)                  ”Daspute, A. A.; Sadhukhan, A.; Tokizawa, M.; Kobayashi, Y.; Panda, S. K.; Koyama, H., Transcriptional regulation of aluminum-tolerance genes in higher plants: clarifying the underlying molecular mechanisms. Front Plant Sci 2017, 8” Please provide full bibliographic data for this reference.

f)                  ”Le Poder, L.; Mercier, C.; Février, L.; Duong, N.; David, P.; Pluchon, S.; Nussaume, L.; Desnos, T., Uncoupling aluminum toxicity from aluminum signals in the STOP1 pathway. Front Plant Sci 2022, 13.”

g)                 Once each word of the journal title is vritten with capital letter, but the other time not. See: ”Theor exp plant phys” (Lines 513-514), ”Crop sci”.

h)                 ”Ranjan, A.; Sinha, R.; Sharma, T. R.; Pattanayak, A.; Singh, A. K., Alleviating aluminum toxicity in plants: implications of 539 reactive oxygen species signalling and crosstalk with other signaling pathways. Physiol. Plant 2021.” Please provide full bibliographic data for this reference.

i)                   Once dot is after the word of the Journal title, but the other time not. See: ”Physiol. Plant”, ”Calif. Agric. Exp. Circ”,   Sci. Rep”, ”Anal. Chim. Acta and ”Front Plant Sci”

j)                   ”Li, Z.; Tang, M.; Hassan, M. J.; Zhang, Y.; Han, L.; Peng, Y., Adaptability to high temperature and stay-green genotypes 544 associated with variations in antioxidant, chlorophyll metabolism, and γ-aminobutyric acid accumulation in creeping 545 bentgrass species. Front Plant Sci 2021, 12.” Please provide full bibliographic data for this reference.

k)                 Journal names should be written with capital letter. See the reference ”Livak, K. J.; Schmittgen, T. D., Analysis of relative gene expression data using real-time quantitative PCR and the 2− ΔΔCT method. methods 2001, 25, (4), 402-408.”

l)                   Once each word of the manuscript title is written with capital letter but the other time not. See: ”Constantine, N. G.; Ries, S. K., Superoxide Dismutases: I. Occurrence in Higher Plants. Plant Physiol 1977, 59, (2), 309-314.” and ”Chance, B.; Maehly, A., Assay of catalases and peroxidases. Methods Enzymol 1955, 2, 764-775.”

m)               ”Cheng, B.; Zhou, M.; Tang, T.; Hassan, M. J.; Zhou, J.; Tan, M.; Li, Z.; Peng, Y., A Trifolium repens flavodoxin‐like quinone reductase 1 (TrFQR1) improves plant adaptability to high temperature associated with oxidative homeostasis and lipids remodeling. Plant J 2023.” Please provide the full bibliographic data for this reference. Please check italicized words in this sentence. All species Latin names should be italicized.

Author Response

Thank you for choosing me as a reviewer of the of the manuscript ID: ijms-2597039 entitled: ,, γ-Aminobutyric acid (GABA) priming alleviates acid-aluminum toxicity and growth retardation by maintaining photosynthetic, antioxidant, and metabolic homeostases in creeping bentgrass” I hope that my comments will help authors to improve their manuscript.

Response: Thank you very much for your review and giving us some good suggestions to improve our manuscript. We have revised the current manuscript according to all suggestions.

Detailed remarks concerning the manuscript:

Q1: I suggest to modify the title of the manuscript to be concise and informative.

Response: Thank you very much for your suggestion. We have modified the title according to suggestion: γ-Aminobutyric acid priming alleviates acid-aluminum toxicity to creeping bentgrass by regulating metabolic homeostasis

Q2: Abstract. Aluminium stress – the doses of stressor used in the should be mentioned. The practical application of the results studies should be provided.

Response: Thank you very much for your suggestion. We have mentioned the dose of stressor used in the Abstract according to suggestion: 5 mmol/L AlCl3·6H2O. The practical application of the results has also been provided according to suggestion: Exogenous GABA priming could improve phytoremediation potential of perennial creeping bentgrass for the restoration of Al-contaminated soils.

Q3: I suggest to divide ”Discussion” section into the subsections corresponded to the results section. The results of the study should be presented on the background of the literature data, not the other way around. The “Discussion” section should be slightly modified as some its parts sounds like introduction.

Response: Thank you very much for your suggestion. The discussion has been divided into subsections just like results section according to suggestion. The “Discussion” section has been slightly modified.

Q4: Materials and method. I suggest to include the subsection concerning statistical analysis.

Response: Thank you very much for your reminding and suggestion. Statistical analysis has been added in the section of Materials and method as 4.6: Statistical analysis was conducted by using SPSS 26.0 (IBM, Armonk, NY, the United States). Least significance test (LSD) and T-test were used to test variance among four treatments at p≤0.05.

Q5: References should be prepared strictly to the guidelines for authors. There are many editorial mistakes that should be improved. Mainly the latest information (bibliography from the latest five years). There is not possible to mention all of them.

There are some examples:

a) All reference should contain full bibliographic data. See: “Hajiboland, R.; Panda, C. K.; Lastochkina, O.; Gavassi, M. A.; Habermann, G.; Pereira, J. F., Aluminum toxicity in plants: present and future. J Plant Growth Regul 2022.”

b) Once the full, but the other time abbreviated Journal names are provided.

c) all the species Latin names should be italicized.

d) ”Iuchi, S.; Koyama, H.; Iuchi, A.; Kobayashi, Y.; Kitabayashi, S.; Kobayashi, Y.; Ikka, T.; Hirayama, T.; Shinozaki, K.; Kobayashi, M., Zinc finger protein STOP1 is critical for proton tolerance in Arabidopsis and coregulates a key gene in aluminum tolerance. PNAS 2007, 104, (23), 9900-9905”. Arabidopsis should be italicized.

e) ”Daspute, A. A.; Sadhukhan, A.; Tokizawa, M.; Kobayashi, Y.; Panda, S. K.; Koyama, H., Transcriptional regulation of aluminum-tolerance genes in higher plants: clarifying the underlying molecular mechanisms. Front Plant Sci 2017, 8” Please provide full bibliographic data for this reference.

f) ”Le Poder, L.; Mercier, C.; Février, L.; Duong, N.; David, P.; Pluchon, S.; Nussaume, L.; Desnos, T., Uncoupling aluminum toxicity from aluminum signals in the STOP1 pathway. Front Plant Sci 2022, 13.”

g) Once each word of the journal title is vritten with capital letter, but the other time not. See: ”Theor exp plant phys” (Lines 513-514), ”Crop sci”.

h) ”Ranjan, A.; Sinha, R.; Sharma, T. R.; Pattanayak, A.; Singh, A. K., Alleviating aluminum toxicity in plants: implications of 539 reactive oxygen species signalling and crosstalk with other signaling pathways. Physiol. Plant 2021.” Please provide full bibliographic data for this reference.

i) Once dot is after the word of the Journal title, but the other time not. See: ”Physiol. Plant”, ”Calif. Agric. Exp. Circ”, ” Sci. Rep”, ”Anal. Chim. Acta” and ”Front Plant Sci”

j) ”Li, Z.; Tang, M.; Hassan, M. J.; Zhang, Y.; Han, L.; Peng, Y., Adaptability to high temperature and stay-green genotypes 544 associated with variations in antioxidant, chlorophyll metabolism, and γ-aminobutyric acid accumulation in creeping 545 bentgrass species. Front Plant Sci 2021, 12.” Please provide full bibliographic data for this reference.

k) Journal names should be written with capital letter. See the reference ”Livak, K. J.; Schmittgen, T. D., Analysis of relative gene expression data using real-time quantitative PCR and the 2− ΔΔCT method. methods 2001, 25, (4), 402-408.”

l) Once each word of the manuscript title is written with capital letter but the other time not. See: ”Constantine, N. G.; Ries, S. K., Superoxide Dismutases: I. Occurrence in Higher Plants. Plant Physiol 1977, 59, (2), 309-314.” and ”Chance, B.; Maehly, A., Assay of catalases and peroxidases. Methods Enzymol 1955, 2, 764-775.”

m) ”Cheng, B.; Zhou, M.; Tang, T.; Hassan, M. J.; Zhou, J.; Tan, M.; Li, Z.; Peng, Y., A Trifolium repens flavodoxin‐like quinone reductase 1 (TrFQR1) improves plant adaptability to high temperature associated with oxidative homeostasis and lipids remodeling. Plant J 2023.” Please provide the full bibliographic data for this reference. Please check italicized words in this sentence. All species Latin names should be italicized.

Response: Thank you very much for your careful review, and we have revised the format of all references strictly according to your suggestions and the guidelines for authors.

Reviewer 2 Report

The paper has some interesting science and good data, but the presentation is very poor and the manuscript needs a great improvement before being suitable for publication.

Figure legends: Please include how did you calculate error bars (statistical error or standard deviation?), the number of replicates per bar (n) and which statistical analysys have you performed (Tukey? Duncan?).

Many graphics are given without the units, for instance Fig. 1C or Fig. 2. Please include the units in the Y axis legend.

Many abbreviations are not defined in the text prior the first use (for instance RWC, chl, Ci, Gs, Pw, etc...) please, include this information.

Figure 3b: Data on Pn, Tr and WUE seems identycal, but apparently the statistical analysis indicates a significative difference. It is a problem of the scale? If this is the case, include these data in a different graphic in order to appreciate the differences. If is a mistake of the statistical analysis, please correct. The same problem, but the other way round, is found in figure 6 were in glucose-6-phospahte and galactinol the bars labeled with a "c" seem quite different. Please correct or explain.

Figure 6: what is the asterisk that appears in the figure? Please include the information in the figure legend.   

English must be revised as there are some grammar mistakes, specifically in the concordance of singular/plural. 

Author Response

The paper has some interesting science and good data, but the presentation is very poor and the manuscript needs a great improvement before being suitable for publication.

Response: Thank you very much for your review and giving us some good suggestions to improve our manuscript. We have revised the current manuscript according to all suggestions. In addition, the language and presentation of the manuscript have been improved by a native English speaker Muhammad Jawad Hassan.

Q1: Figure legends: Please include how did you calculate error bars (statistical error or standard deviation?), the number of replicates per bar (n) and which statistical analysis have you performed (Tukey? Duncan?).

Response: Thank you very much for your reminding and good question. Statistical analysis has been added in the section of Materials and method as 4.6: Statistical analysis was conducted by using SPSS 26.0 based on Tukey test (IBM, Armonk, NY, the United States). Least significance test (LSD) and T-test were used to test variance of different treatments at p≤0.05. The “Vertical bars above columns indicate standard error (n=3)” has been added in each figure legend.

Q2: Many graphics are given without the units, for instance Fig. 1C or Fig. 2. Please include the units in the Y axis legend.

Response: Thank you very much for your reminding and suggestion, and the unit of Fig. 1C (relative growth rate) is the ‘mg g-1 d-1’, the unit of Fig. 2A (relative water content) is the ‘%’, and the unit of Fig.3A (chl a, chl b, total chl content) is the ‘mg g-1 DW’. The units have been added in all graphics if these parameters were presented by using the units.

Q3: Many abbreviations are not defined in the text prior the first use (for instance RWC, chl, Ci, Gs, Pw, etc...) please, include this information.

Response: Thank you very much for your reminding and suggestion, and we have added full names when these abbreviations were used first time throughout the entire manuscript.

Q4: Figure 3b: Data on Pn, Tr and WUE seems identycal, but apparently the statistical analysis indicates a significative difference. It is a problem of the scale? If this is the case, include these data in a different graphic in order to appreciate the differences. If is a mistake of the statistical analysis, please correct. The same problem, but the other way round, is found in figure 6 were in glucose-6-phospahte and galactinol the bars labeled with a "c" seem quite different. Please correct or explain.

Response: Thank you very much for your careful review and question. Yes, data on Pn, Tr and WUE seems identycal because of a problem of the scale. The numerical differences between Ci and other parameters (Pn, Tr and WUE) in the figure 3b were too large to make other parameters seems identical. We remade the figure 3b, so that differences of other parameters could be obviously observed now. In addition, the statistical analysis and significance for glucose-6-phospahte and galactinol were correct. The reason why the bars labeled for the two treatments (Al and Al+GABA) of glucose-6-phospahte and galactinol with a “c” seem quite different, because numerical values of treatments C and C+GABA were more than ten times higher than those of Al and Al+GABA, resulting in no significant difference between Al and Al+GABA if we calculated differences based on four treatments (C, C+GABA, Al, and Al+GABA). However, significant difference was detected between Al and Al+GABA when we only calculated differences based on these two treatments (Al and Al+GABA). As you can see from the Y-axis of figure 6C or 6D, there was or were one or two separations to make all data can be obviously observed. We have also added the explanation in the legend of figure 6 according to suggestion: The “*” represents a significant difference between two treatments (Al and Al+GABA).

Q5: Figure 6: what is the asterisk that appears in the figure? Please include the information in the figure legend.

Response: Thank you very much for your question. Asterisks indicate significant differences between two treatments, and We have added the explanation in the legend of figure 6 according to suggestion: The “*” represents a significant difference between two treatments (Al and Al+GABA).

Q6: English must be revised as there are some grammar mistakes, specifically in the concordance of singular/plural.

Response: Thank you very much for your suggestion. The manuscript has been modified and improved according to your suggestion.

Round 2

Reviewer 1 Report

Dear Editor of the International Journal of Molecular Sciences,

Thank you once again for choosing me as reviewer of the manuscript ijms-2597039 entitled “γ-Aminobutyric acid (GABA) priming alleviates acid-aluminum toxicity and growth retardation by maintaining photosynthetic, antioxidant, and metabolic homeostases in creeping bentgrass”. I would also to thank Authors for their response to the comments and efforts in careful correction of the manuscript. Now the papers may be accepted.

Reviewer 2 Report

Authors have made a great improvement in the manuscript

I recommedn publication

English is fine.